# Estradiol Induces Epithelial to Mesenchymal Transition of Human Glioblastoma Cells

**DOI:** 10.3390/cells9091930

**Published:** 2020-08-21

**Authors:** Ana M. Hernández-Vega, Aylin Del Moral-Morales, Carmen J. Zamora-Sánchez, Ana G. Piña-Medina, Aliesha González-Arenas, Ignacio Camacho-Arroyo

**Affiliations:** 1Unidad de Investigación en Reproducción Humana, Instituto Nacional de Perinatología-Facultad de Química, Universidad Nacional Autónoma de México, México City CP 11000, Mexico; anahdzvg@gmail.com (A.M.H.-V.); aylindmm@gmail.com (A.D.M.-M.); carmenjaninzamora@comunidad.unam.mx (C.J.Z.-S.); 2Departamento de Biología, Facultad de Química, Universidad Nacional Autónoma de México, México City CP 04510, Mexico; a.gabriela.pime@gmail.com; 3Departamento de Medicina Genómica y Toxicología Ambiental, Instituto de Investigaciones Biomédicas, Universidad Nacional Autónoma de México, México City CP 04510, Mexico; alieshag@iibiomedicas.unam.mx

**Keywords:** epithelial–mesenchymal transition (EMT), glioblastoma multiforme (GBM), 17β-estradiol (E2), estrogen receptors (ERs)

## Abstract

The mesenchymal phenotype of glioblastoma multiforme (GBM), the most frequent and malignant brain tumor, is associated with the worst prognosis. The epithelial–mesenchymal transition (EMT) is a cell plasticity mechanism involved in GBM malignancy. In this study, we determined 17β-estradiol (E2)-induced EMT by changes in cell morphology, expression of EMT markers, and cell migration and invasion assays in human GBM-derived cell lines. E2 (10 nM) modified the shape and size of GBM cells due to a reorganization of actin filaments. We evaluated EMT markers expression by RT-qPCR, Western blot, and immunofluorescence.We found that E2 upregulated the expression of the mesenchymal markers, vimentin, and N-cadherin. Scratch and transwell assays showed that E2 increased migration and invasion of GBM cells. The estrogen receptor-α (ER-α)-selective agonist 4,4’,4’’-(4-propyl-[1H]-pyrazole-1,3,5-triyl)trisphenol (PPT, 10 nM) affected similarly to E2 in terms of the expression of EMT markers and cell migration, and the treatment with the ER-α antagonist methyl-piperidino-pyrazole (MPP, 1 μM) blocked E2 and PPT effects. ER-β-selective agonist diarylpropionitrile (DNP, 10 nM) and antagonist 4-[2-phenyl-5,7-bis(trifluoromethyl)pyrazole[1,5-a]pyrimidin-3-yl]phenol (PHTPP, 1 μM) showed no effects on EMT marker expression. These data suggest that E2 induces EMT activation through ER-α in human GBM-derived cells.

## 1. Introduction

Malignant tumors of the central nervous system (CNS) are among the cancers with the worst prognosis. Glioblastoma multiforme (GBM) comprises approximately half of the malignant primary brain tumors and causes 3–4% of cancer-related deaths [1]. The World Health Organization defines GBM as a grade IV astrocytoma tumor characterized by uncontrolled proliferation, necrosis propensity, angiogenesis, deep infiltration, apoptosis resistance, genomic instability, and extensive heterogeneity at the cellular and molecular levels [2,3]. The Cancer Genome Atlas (TCGA) network identified four molecular subtypes of GBM on the basis of the gene expression profile of neural progenitor cells (proneural, PN), neurons (neural, N), proliferative cells with activation of the tyrosine kinase receptor (classical, CL), and mesenchymal tissue (mesenchymal, MES) [4]. The mesenchymal phenotype of GBM tends to have the worst survival rates compared to the other subtypes, and it is associated with a highly invasive behavior [5,6,7].

Epithelial-to-mesenchymal transition (EMT) is a mechanism of cellular plasticity that regulates a set of transient states between the epithelial and mesenchymal phenotype. During EMT, epithelial cells lose their junctions with other cells and the apicobasal polarity while they acquire a mesenchymal phenotype with migratory and invasive properties [8]. EMT is a highly dynamic and transient mechanism induced by diverse signals and orchestrated by EMT-inducing transcription factors (EMT-TFs), which act in close association with the epigenetic machinery by repressing epithelial genes and activating mesenchymal genes [9]. The reverse process is the mesenchymal–epithelial transition (MET) [10]. EMT is essential in diverse physiological and pathological processes [11]. This mechanism is associated with embryogenesis [12,13,14], heart regeneration [15], wound healing, fibrosis, and organ repair [16]. In tumor cells, the activation of EMT-TFs promotes the mechanisms of migration, invasion, metastasis, apoptosis inhibition, resistance to radio- and chemotherapy, as well as maintenance of the plasticity of cancer stem cells [17].

Although EMT is typical in epithelial tumors, evidence suggests that EMT-TFs also lead to a gain in mesenchymal properties and the promotion of malignancy of non-epithelial tumors, including brain tumors, hematopoietic malignancies, and sarcomas [18,19]. Currently, the classic description of EMT as a process of change between two alternative states (epithelial and mesenchymal) has been replaced by a new concept of cellular plasticity and transient states, which proposes that cells move through a spectrum of various intermediate phases, which means that cells can carry out partial EMT programs [20]. Several studies have shown the role of EMT in GBM progression. Large-scale expression analysis of 85 highly diffuse glioma tumors revealed a set of genes associated with mesenchymal tissue overexpressed in GBM biopsies [21]. Tso et al. showed that a subset of primary GBM tumors expresses cellular and molecular markers associated with mesenchymal stem cells [22]. Then, the definition of the GBM mesenchymal subtype convincingly showed the clinical importance of the EMT program in tumor diagnosis and treatment [4,5]. Molecular profile analysis of the four GBM subtypes demonstrated that the mesenchymal subtype, unlike the other subtypes, presents the molecular characteristics of EMT [23].

Determination of the complete molecular network of the EMT program, as well as the fundamental mechanisms necessary to activate it, could provide new therapeutic approaches for GBM treatment. Autocrine and paracrine interactions within the GBM microenvironment induce EMT through intracellular signaling pathways that activate EMT-TFs. Although different studies have described several signaling pathways that induce EMT in GBM, the role of the different factors within the tumor microenvironment, as well as all the interactions that coordinate this cellular program, is still not understood.

Sex steroid hormones such as estrogens participate in a wide variety of functions throughout the nervous system. These hormones are mainly synthesized in the gonads and the adrenal glands, but they can also be produced de novo within the brain [24,25]. Estrogens include estrone (E1), 17β-estradiol (E2), and estriol (E3). E2 is involved in many brain functions, such as brain development during sexual differentiation [26], differentiation of neurons and glial cells [27,28], and regulation of neurite growth and synaptic patterns [29,30], and it interacts with the glutamatergic, dopaminergic, and serotonergic neurotransmission pathways that influence the generation of memory, learning, and emotional state [31,32,33]. Estrogens act by binding specific intracellular and membrane receptors. There are two estrogen-specific intracellular receptor subtypes, estrogen receptors α and β (ERα and ERβ), which are ligand-activated transcription factors that directly regulate gene expression. Moreover, these receptors are associated with the plasma membrane, where they activate intracellular signaling pathways [34].

E2 concentrations, as well as ERs expression and activity, are determinant in the malignant progression of tumors growing in estrogen-sensitive tissues [35,36,37,38]. On the basis of these studies, ER-α promotes cell proliferation, whereas ER-β has anti-proliferative effects [39]. ER expression status in GBM is controversial. Some studies have reported the absence of ER in GBM [40,41], while other researchers have determined that ER expression varies according to malignancy degree, suggesting that these receptors are involved in GBM malignant progression [42,43,44,45,46,47,48,49,50]. ER subtypes have shown different effects in GBMs. E2 and 4,4’,4’’-(4-propyl-[1H]-pyrazole-1,3,5-triyl)trisphenol (PPT), a selective agonist of ER-α, increased the number of cells derived from human GBM [45], while ER-β-specific agonists decreased GBM cell proliferation [44]. However, the molecular mechanisms of E2 related to GBM malignant progression are still unclear.

E2-promoted signaling is known to be related to EMT induction in estrogen-responsive tissues. In ovarian and prostate cancer, E2 treatment induces ER-α-dependent EMT, while receptor silencing inhibits EMT [51,52,53]. Nevertheless, loss of ER-α expression in breast and endometrial cancer promotes morphological changes, motility, and improved invasion, as well as increased expression of EMT markers [54,55,56,57,58]. These investigations demonstrate the importance of specific cell context in the E2-induced EMT. However, E2 involvement in the EMT program in GBM is unknown.

To increase the knowledge regarding the EMT program of GBM, in this study, we investigated the participation of E2 on EMT induction in human GBM-derived cells expressing both ER subtypes. Our results showed that the treatment with E2 (10 nM) promoted: (1) changes in cell morphology and the structure of the actin cytoskeleton, (2) increased expression of mesenchymal markers such as vimentin and N-cadherin, and (3) increased migratory and invasive capacity of GBM cells. These effects were dependent on ER-α, since the treatment with its agonist, PPT (10 nM), produced similar results to E2, while the treatment with its antagonist methyl-piperidino-pyrazole (MPP, 1 μM), blocked the effects of E2 and PPT.

## 2. Materials and Methods

### 2.1. TCGA Data Analysis

Ribonucleic acid sequencing (RNA-Seq) counts were obtained from low-grade gliomas (LGG, *n* = 167) and glioblastoma (GBM, *n* = 155) projects of The Cancer Genome Atlas (TCGA) repository (https://portal.gdc.cancer.gov/). The data were downloaded and processed using TCGAbiolinks package version 2.12.6 for R [59]. Additionally, expression profiles were obtained from healthy brain cortex samples (*n* = 249) in the GTEx database (https://gtexportal.org/home/). Data were normalized by DESeq2 version 1.22.2 [60] and plotted. TCGA_analyse_survival utility from the TCGAbiolinks package for R performed survival analysis.

### 2.2. Cell Cultures

Human GBM-derived cell lines U87, U251, T98, and LN229 (American Type Culture Collection, ATCC, Manassas, VA, USA) were cultivated in Dulbecco’s modified Eagle’s medium (DMEM, L0107-500) high glucose supplemented with 10% fetal bovine serum (FBS; S1650), 1.0 mM pyruvate (L0642-100), 1.0 mM antibiotic (streptomycin 10 g/L; penicillin G 6.028 g/L; and amphotericin B 0.025 g/L, L0010), and 0.1 mM non-essential amino acids (X0557-100, Biowest, Nuaillé, PDL, France). Cell cultures were maintained at 37 °C in a humidified atmosphere with 5% CO_2_. At 60% confluence (24 h before treatments), cells were culture in DMEM no phenol red (ME-019 Thermo Fisher Scientific, Waltham, MA, USA) supplemented with 10% charcoal/dextran-treated FBS (SH30068.03, Thermo Fisher Scientific), 1.0 mM pyruvate, 1.0 mM antibiotics, and 0.1 mM non-essential amino acids. When indicated, cells were treated with E2 (10 nM, E4389, Sigma-Aldrich, St. Louis, MO, USA), ER-α-selective agonist PPT (10 nM, 1426, Tocris, Bristol, UK, England), ER-β-selective agonist diarylpropionitrile (DNP, 10 nM, 1494, Tocris), ER-α-selective antagonist MPP (1 μM, 1991, Tocris), and ER-β-selective antagonist 4-[2-phenyl-5,7-bis(trifluoromethyl)pyrazole[1,5-a]pyrimidin-3-yl]phenol (PHTPP, 1 μM, 2662, Tocris). In combined treatments, antagonists MPP and PHTPP were added 2 h before the addition of agonist.

### 2.3. Cell Morphology Analysis

The epithelial phenotype is characterized by a polygonal shape, while the mesenchymal phenotype is spindle-shaped. Therefore, the geometric characteristics of both phenotypes differ from each other. Geometric characteristics can be quantified using high-performance software for the analysis of cell images [61,62,63,64]. The morphological changes of the U251, U87, T98G, and LN229 cells treated with vehicle and E2 at 0, 48, and 72 h were determined by phase contrast microscopy (IX71, inverted microscope Olympus, Shinjuku, TY, Japan), digitally capturing six arbitrary fields with a 400X magnification for each of the treatments. Adobe Photoshop CS6 software (Adobe Systems Inc., San Jose, CA, USA) was used to process the background correction and illumination of the captured images. Subsequently, the orientation, shape, and position of each of the cells in each image was determined to segment them with the Image-Pro software 10.0.6 (Media Cybernetics Inc., Rockville, MD, USA), which has automated algorithms to identify, separate, and quantify each of the cells that appear in the image. This quantification allows the extraction of various geometric characteristics that determine morphological parameters of the cells segmented in the two-dimensional plane.

### 2.4. RT-qPCR

Total RNA was extracted from cells by guanidine–thiocyanate–phenol–chloroform method with TRIzol LS Reagent (10296028, Thermo Fisher Scientific, Waltham, MA, USA), following the supplier’s protocol, and was measured by spectrophotometry (Nanodrop 2000 spectrophotometer, Thermo Fisher Scientific). RNA integrity was checked by electrophoresis with 1.5% agarose gel in Tris-Borate-ethylenediaminetetraacetic acid (EDTA) buffer (TBE: 89 mM Tris, 89 mM boric acid, 2.0 mM EDTA (pH 8.3)) detected by fluorescence with GreenSafe (MB13201, NZYTech, Lisboa, PT, Portugal). Human astrocyte RNA was purchased from ScienCell Research Laboratories (1805, Carlsbad, CA, USA). Moloney Murine Leukemia Virus Reverse Transcriptase (M-MLV RT, 28025013, Thermo Fisher Scientific) was used to obtain the complementary DNA (cDNA) from one microgram of extracted RNA following the protocol recommended by the provider. Gene expression relative to the 18S ribosomal RNA (rRNA) gene was quantified through the quantitative polymerase chain reaction (qPCR) using standardized primers for each gene: ESR1 (estrogen receptor 1/α) (FW-5′-agcaccctgaagtctctgga-3′, RV-5′-gatgtgggagaggatgagga-3′); ESR2 (estrogen receptor 2/β) (FW-5′-aagaagattcccggctttgt-3′, RV-5′-tctacgcatttcccctcatc-3′); VIM (vimentin) (FW-5′-ggaccagctaaccaacgaca-3′, RV-5′-aaggtcaagacgtgccagag-3′); CDH2 (cadherin-2/N-cadherin) (FW-5′-ctggagacattggggacttc-3′, RV-5′-gagccactgccttcatagt-3′); TJP1 (tinght junction protein 1/zonula occludens 1 (ZO-1)) (FW-5′-gccattcccgaaggagttga-3′, RV-5′-atcacagtgtggtaagcg-3′); rRNA18S (FW-5′-agtgaaactgcgaatggctc-3′, RV-5′-ctgaccgggttggttttgat-3′). FastStart DNA Master SYBR Green I kit (12239264001, Roche, Basel, Switzerland) was used to perform gene amplification in a LightCycler 2.0 instrument (03531414001, Roche). Relative expression was quantified by the comparative 2^ΔΔCt^ method [65,66].

### 2.5. Western Blot

U251, U87, T98G, and LN229 cells were detached from culture plates using cold phosphate-buffered saline (PBS: 137 mM NaCl, 2.7 mM KCl, 10.0 mM Na_2_HPO_4_, 1.8 mM KH_2_PO_4_ (pH 7.4)) and cell scraper. The pellet obtained from the centrifuged cells at 45 × *g* for 3 min were lysed with radioimmunoprecipitation assay buffer (RIPA: 50 nM Tris-HCl, 150 nM NaCl, 1% Triton X-100, 0.1% sodium dodecyl sulfate (SDS) (pH 8.0)) supplemented with protease inhibitor cocktail (P8340, Sigma-Aldrich, St. Louis, MO, USA). Total proteins were extracted by centrifugation at 20,817 × *g* at 4 °C for 15 min and quantified by spectrophotometry (NanoDrop 2000 spectrophotometer, Thermo Fisher Scientific, Waltham, MA, USA) using the Pierce 660 nm protein assay reagent (22660, Thermo Fisher Scientific). Thirty µg of total protein was separated on 10% SDS-polyacrylamide gel electrophoresis (PAGE) at 80 V for 4 h and then transferred to a polyvinylidene fluoride (PVDF) membrane (IPVH00010, Merck, Kenilworth, NY, USA) at 20 V in semidry conditions at room temperature for 45 min. Membranes were blocked with 5% bovine serum albumin (BSA, A9418, Sigma-Aldrich ) in Tris-buffered saline-Tween (TBST: 150 mM NaCl, 50 mM Tris-HCl, 0.1% Tween (pH 7.6)) with constant agitation at 37 °C for 2 h, and then incubated with primary antibodies: anti-ERα (2 μg/mL, rabbit polyclonal, ab3575, Abcam, Cambridge, UK, England), anti-ERβ (0.4 μg/mL, mouse monoclonal 1531: sc-53494), anti-ZO-1 (0.6 μg/mL, rat monoclonal R40.76: sc-33725), anti-N-cadherin (0.8 μg/mL, mouse monoclonal D-4: sc-8424), anti-vimentin (0.4 μg/mL, mouse monoclonal V9: sc-6260), and α-tubulin (0.4 μg/mL, mouse monoclonal A-6: sc-398103) (Santa Cruz Biotechnology, Dallas, TX, USA), diluted with 5% BSA in TBST at 4 °C for 48 h. Subsequently, the membranes were washed with TBST three times every 5 min and incubated with secondary antibodies conjugated to horseradish peroxidase (HRP): anti-rabbit (0.06 μg/mL, goat polyclonal Immunoglobulin G (IgG) containing two heavy chains (H) and two light chains (L) (H+L), 65-6120; Thermo Fisher Scientific), anti-mouse (0.013 μg/mL, purified recombinant mouse IgGκ light chain: sc-516102; Santa Cruz Biotechnology), and anti-rat (0.06 μg/mL, goat polyclonal IgG (H+L), ab97057, Abcam) at room temperature and constant agitation for 45 min, and again washed with TBST three times every 5 min. Finally, Super Signal West Femto Maximum Sensitivity Substrate reagent (34096, Thermo Fisher Scientific) was incubated in the membranes, and immunoreactive bands were detected by chemiluminescence exposing blots to Kodak Biomax Light Film (Z370371, Sigma-Aldrich) captured by a digital camera of 14.1 megapixels (SD1400IS, Canon Inc., Ota, TY, Japan). ImageJ software (1.52u, National Institutes of Health, NIH, Bethesda, MD, USA) performed the densitometric analysis of blot images.

### 2.6. Immunofluorescence

U251 and U87 cells were fixed with 4% paraformaldehyde (4% PFA) at room temperature for 20 min, washed with PBS, and then incubated in permeabilizing blocking solution (1% BSA, 1% glycine, 0.2% Triton X-100, diluted in PBS) at room temperature for 90 min. Subsequently, cells were incubated with primary antibodies: anti-Actin (4 μg/mL, goat polyclonal C-11, sc-1615), anti-ZO-1 (8 μg/mL, rat monoclonal R40.76: sc-33725), anti-N-cadherin (8 μg/mL, mouse monoclonal D-4: sc-8424), and anti-vimentin (4 μg/mL, mouse monoclonal V9: sc-6260) (Santa Cruz Biotechnology, Dallas, TX, USA) at 4 °C overnight, and then rinsed three times every 5 min with PBST (PBS with 0.05% Tween). Later, cells were incubated with secondary antibodies: anti-mouse (4 μg/mL, goat polyclonal IgG (H+L) Alexa Fluor 488: A11001, Thermo Fisher Scientific, Waltham, MA, USA), anti-rat (4 μg/mL, goat polyclonal IgG (H+L) Alexa Fluor 488: ab150157, Abcam, Cambridge, UK, England), and anti-goat (8 μg/mL, donkey IgG-FITC: sc-2024, Santa Cruz Biotechnology) at room temperature for 90 min, and again rinsed with PBST three times every 5 min. Nuclei were stained with Hoechst 33,342 (1 mg/mL, 62249, Thermo Fisher Scientific) at room temperature for 7 min and rinsed three times every 5 min with PBST. Finally, cells were coverslipped with mounting medium (18606-20, Polysciences, Warrington, PA, USA) and visualized by fluorescence microscopy (Bx43, light microscope, Olympus, Shinjuku, TY, Japan), digitally capturing six arbitrary fields with a 400× magnification. Fluorescence density was measured as integrated density from the *Analyze* menu of ImageJ software.

### 2.7. Migration Assay

Wound healing assays were performed to determine the migratory capacity of cells. U251 and U87 cells grew in DMEM high glucose supplemented until reaching 70% confluence. Then, the medium was changed to DMEM no phenol red supplemented 10% charcoal/dextran-treated FBS, 1.0 mM pyruvate, 1.0 mM antibiotics, and 0.1 mM non-essential amino acids, and incubated at 37 °C in a humidified atmosphere with 5% CO_2_. Upon 90% confluence, a scratch was made using a 200 μL pipette tip. Cells floating were rinsed with PBS and DMEM no phenol red-supplemented 10% charcoal/dextran treated-FBS, 1.0 mM pyruvate, 1.0 mM antibiotics, and 0.1 mM non-essential amino acids were added again. One hour before adding the experimental treatments, we incubated the cells with cytosine β-D-arabinofuranoside hydrochloride (10 μM, Ara-C, C1768, Sigma-Aldrich, St. Louis, MO, USA), a selective inhibitor of DNA synthesis. Images of the wound area captured at 100× magnification with an Infinity 1-2C camera (Lumenera, Otawa, ON, Canada) connected to an inverted microscope (CKX41, Olympus, Shinjuku, TY, Japan) at 0, 12, and 24 h of treatment were analyzed using the MRI Wound Healing Tool plugins of Image J software.

### 2.8. Invasion Assay

Transwell assay determined the invasion potential of cells. Transwell inserts with 10 μm membrane thickness and 8 μm pore size (3422, Corning, Corning, NY, USA) were placed in 24-well plates, and each well was covered with 50 μL of ECM Gel from Engelbreth-Holm-Swarm murine sarcoma (2 mg/mL, matrigel E1270, Sigma-Aldrich, St. Louis, MO, USA) diluted in DMEM no phenol red without supplement, and immediately incubated at 37 °C for 2 hours. Then, 15,000 U87 cells or 10,000 U251 cells suspended in 150 μL DMEM no phenol red and without supplement with 10 μM Ara-C and treatments (vehicle or E2 10 nM) were added to the upper insert, while the lower wells were filled with 500 μL DMEM supplemented with 10% FBS as a chemoattractant were incubated in a humidified atmosphere with 5% CO_2_ at 37 °C for 24 h. Transwell inserts were rinsed with PBS, fixed with 4% PFA for 20 min, and stained with 0.1% crystal violet dye for an additional 20 min. Inserts were washed three times with PBS for each 15 min in order to remove excess dye. Finally, images of invasive cells captured at 100× magnification with an Infinity 1-2C camera (Lumenera, Otawa, ON, Canada) connected to an inverted microscope (CKX41, Olympus, Shinjuku, TY, Japan) were analyzed using the Cell Counter plugin in the ImageJ software.

### 2.9. Statistical Analysis

Data were analyzed and plotted with the GraphPad Prism 5.0 software (GraphPad, San Diego, CA, USA). Statistical analysis between comparable groups was performed using a one-way ANOVA with a Tukey post hoc-test. Time course analysis was performed using a two-way ANOVA test followed by Bonferroni post-test to compare replicate means by row. Values of *p* < 0.05 were considered statistically significant. Plotted data are representative of three independent experiments for each treatment.

## 3. Results

### 3.1. Differential Expression of ERα and ERβ Subtypes in Human GBM-Derived Cells

We evaluated the mRNA expression levels of ESR1 (ER-α) and ESR2 (ER-β) genes in astrocytoma samples with different histological grades from the data obtained of the low-grade gliomas (LGG) and glioblastoma (GBM) projects from the TCGA repository, as well as samples of healthy cerebral cortex in the GTEx database. Low ESR1 expression levels were observed in GBM and LGG when compared to healthy tissue. A slight but significant increase was found in ESR1 expression in GBM as compared with LGG. These data highlight two critical points: (1) ESR1 expression was lower in gliomas compared to healthy tissue, (2) but also was higher in GBM compared to LGG, suggesting an important oncogenic role of ER-α in development of low- and high-grade gliomas. In contrast, ESR2 expression levels were higher in GBM as compared with LGG and healthy tissue (Figure 1A). Next, we compared ESR1 and ESR2 expression among the four GBM subtypes defined by Verhaak et al. [4]. The mesenchymal subtype showed higher levels of ESR1 mRNA expression compared to the classical and neural subtypes, without significant differences when compared with the proneural subtype. ESR2 expression in the mesenchymal subtype showed a tendency to be the highest, although it was only significantly higher when compared with the neural subtype (*p* < 0.05) (Figure 1B). It is interesting to highlight that our results showed that expression of both ER subtypes was enriched in the mesenchymal subtype. Analysis of expression in cell lines showed a similar trend to the TCGA data: the expression of both ESR1 and ESR2 in four cell lines derived from human GBM (U251, U87, T98G, and LN229) was found to be lower compared to the expression of normal human astrocytes (NHA). Among GBM cells, the ESR1 gene was expressed in a higher proportion in U251 cells, while ESR2 had a higher expression in U87 cells (Figure 1C). We evaluated the expression of both ERs subtypes at the protein level in GBM cells, and a higher content of ER-α than that of ER-β was observed in all cell lines. Moreover, we found two isoforms of ER-β (ER-β1 and ER-β5) expressed in GBM cells; ER-β1 was more abundant than ER-β5 (Figure 1D). An analysis of the clinical outcome of ER expression in GBM patients showed that the higher expression of ER-α and ER-β was correlated with a poor prognosis. Therefore, patients with a low expression of both ER subtypes live longer than those with higher levels of expression (Figure 1E). Importantly, although lower ER-α expression was observed in GBM TCGA data compared to healthy tissue, survival analysis showed that high ER-α expression is a poor prognostic factor for the patients, which suggests that ER-α expression levels in GBM may not always be proportional to its oncogenic activity. These results suggest that ER-α and ER-β expression differentially changes among healthy tissue, LGG, and GBM, both in vivo and in vitro, and it also varies among GBM subtypes.

### 3.2. Changes in Cell Morphology During E2-Induced EMT

Morphological changes associated with EMT in U251 and U87 cells were evaluated after E2 (10 nM) treatment. At the beginning of the treatments, U251 cells presented a typical star-like morphology, and U87 cells a polygonal shape. Interestingly, the cells treated with E2 showed a spindle-shape and the typical features of mesenchymal cells at 48 and 72 h (Figure 2). Table 1 details the geometric parameters quantified in this work. Values close to the unity of the circularity and box XY (width/height) measurements are characteristic of a polygonal shape, while a high aspect (major/minor axis) and perimeter denote a fusiform shape. Plots show that E2 decreased circularity and box XY, while increased aspect and perimeter (Figure 2). This effect on the cellular morphology was consistent in T98G and LN229 cells (Appendix A). These results show that E2 promotes morphological changes associated with EMT.

### 3.3. E2-Induced Reorganization of Actin Filaments

To determine whether the morphological changes observed above were related to changes in the arrangement of actin filaments, we performed immunofluorescence assays in U251 and U87 cells. In U251 cells treated with vehicle, the actin filaments were predominantly organized into bundles of dense reticulated mesh, characteristic of cortical actin. In contrast, in cells treated with E2 (10 nM), the actin filaments assembled in parallel along the ventral surface of the cell, forming long projections towards the leading edge, which in the extreme showed focal sites with a high concentration of actin (Figure 3). In U87 cells, we observed a higher proportion of concentrated actin focal points, both in vehicle and E2 treated cells. However, cells incubated with E2 showed long parallel filament projections with a high concentration of actin around the edge (Figure 3). Thereby, the morphological changes induced by E2 in GBM cells were related to a reorganization of the actin filaments.

### 3.4. E2 Regulated EMT Marker Expression

We analyzed the effects of E2 on EMT marker expression and distribution in GBM cells by RT-qPCR, Western blot, and immunofluorescence. We evaluated the peripheral membrane protein zonula occludens 1 (ZO-1, encoded by the TJP1 gene) as an epithelial marker, and the evaluated mesenchymal phenotype markers were N-cadherin (encoded by the CDH2 gene) and vimentin (encoded by the VIM gene). In U251 cells, E2 increased TJP1 expression only at 48 h, CDH2 expression from 24 to 72 h, and VIM expression from 48 h (Figure 4A). Importantly, ZO-1 protein content showed no changes due to E2 treatment, while the hormone increased the content of N-cadherin and vimentin proteins at 72 h (Figure 4B). E2 effects on U87 cells were like those of U251 cells. E2 upregulated TJP1 and ZO-1 expression from 48 h, CDH2 expression from 24 h, and N-cadherin expression from 48 h; VIM expression increased at 24 h, and decreased at 72 h, while the vimentin protein content increased at 72 h (Figure 4C,D).

The analysis of EMT markers by immunofluorescence showed that in U251 and U87 cells treated with the vehicle, ZO-1, and N-cadherin proteins were expressed in localized regions of the plasmatic membrane, particularly at cell-binding sites. In contrast, in E2-treated cells, these proteins were shown along the entire cell surface, especially in long projections at the cell ends (Figure 5A,B). Vimentin filaments formed a network within the cytoplasm in cells without E2, whereas in E2-incubated cells, vimentin filaments were arranged in parallel along the ventral surface of the cell, particularly at the borders, similar to actin filaments (Figure 5A,B). Overall, the fluorescence intensity of EMT markers significantly increased in E2-treated U251 and U87 cells (Figure 5C). These results show that E2 induces EMT marker expression and its redistribution in GBM cells.

Furthermore, the increase in both epithelial and mesenchymal markers suggests that E2 promotes the induction of a partial EMT, which expresses both phenotypes. Nevertheless, the increase in ZO-1 in U251 cells was not as evident as in U87 cells, and thus these results are not sufficient to determine the status of E2-induced EMT. However, these results showed that E2 significantly induced the expression of mesenchymal markers, promoting the mesenchymal phenotype of cells derived from GBM.

### 3.5. E2 Promoted Migration and Invasion of Human GBM-Derived Cells

We evaluated cell migration by wound healing assay, and we observed that E2-treated U251 and U87 cells showed a higher migratory capacity by rapidly closing the wound compared to cells without E2 (Figure 6A,B). We also evaluated invasive capacity through transwell assay. E2 increased the number of invading U251 and U87 cells as compared with the vehicle (Figure 6C,D). These data show that E2, in addition to changing cell morphology and regulating EMT marker expression, also increased the migratory and invasive capacity of GBM-derived cells.

### 3.6. ER-α Mediated E2 Effects on EMT

To determine the intracellular receptor subtype involved in E2 effects, we used specific agonists and antagonists’ ER subtypes and assessed the expression of EMT markers. PPT, a selective ER-α agonist, increased TJP1, CDH2, and VIM gene expression in a similar way to E2 in U251 and U87 cells, and ER-α antagonist MPP blocked E2 effects in both cell types (Figure 7A,B). Treatment with antagonist alone did not show a significant effect on the expression of EMT markers, consistent with characterization of the effects of MPP in vitro, which showed that the antagonist does not behave as a partial or inverse agonist when administered in absence of an agonist. Treatments with the ER-β-selective agonist DNP and the antagonist PHTPP did not show any significant statistical effect on the regulation of EMT marker expression in either U251 or U87 cells (Figure 7A,B). These data suggest that E2 regulates EMT marker expression through the ER-α subtype. To functionally assess the role of ER-α in the EMT process, we performed wound healing assays using PPT and MPP. PPT-treated U251 cells rapidly closed the wound compared to cells treated with the vehicle, while the antagonist MPP blocked the PPT effect. Similarly, PPT increased the wound closure rate of U87 cells, and MPP blocked the agonist effect (Figure 8).

## 4. Discussion

The present study provides evidence of E2 effects on EMT-related molecular and cellular processes in human GBM-derived cells. EMT comprises a set of states between the epithelial and mesenchymal phenotypes, and its activation could be closely related to the high degree of phenotypic heterogeneity of GBM.

GBM is a highly heterogeneous tumor at both the molecular and cellular levels. The TCGA determined the existence of four main subtypes with different molecular expression profiles [4], and some studies have shown the simultaneous presence of these subtypes within the same tumor [67,68]. GBM cells with proneural and mesenchymal expression are the most consistent subtypes in the literature. The proneural subtype is related to a more favorable prognosis, and the mesenchymal subtype tends to have the worst survival rate [5,69,70]. It has been shown that recurring GBM tumors that initially showed proneural expression presented mesenchymal expression profile after radiotherapy and chemotherapy [6,71,72], which has led to the proposal of a proneural–mesenchymal transition (PMT), whose molecular events are equivalent to those of EMT [7,73]. The transition between the two molecular subtypes is closely related to an enrichment of cells of the immune system within the GBM tumor microenvironment, which activate various signaling pathways that promote PMT/EMT [74,75,76]. The data shown in these studies highlight the importance of the factors found within the tumor microenvironment that promote phenotypic transitions between GBM subtypes. Immune system cells produce chemokines, cytokines, growth and angiogenic factors, immunosuppressive molecules, and extracellular matrix-modifying enzymes, which make the surroundings favorable for tumor progression [77]. Among the many factors found within the GBM tumor microenvironment, in this work, we focused on E2, which is produced by microglial cells, astrocytes, and GBM cells [78,79,80,81]. In this study, we evaluated E2 effects on the mesenchymal transition of human GBM-derived cells.

Both ER subtypes are predominantly expressed through healthy CNS; however, in human astrocytomas, both ER-α [45,47,48,49] and ER-β [44] expression decreases as the grade of tumor malignancy increases. Therefore, different researchers have proposed that ER expression may be reduced or lost during tumor development, although it is not clear if this represents a cause or consequence of tumor development. Much remains to be investigated on this topic, since determining the mechanisms underlying ER decrease during the development of gliomas could better understand the malignant tumor progression. Our results regarding the decrease of ER-α expression in gliomas compared with healthy tissue show agreement with that observed in other investigations. Likewise, we showed that ER-α subtype expression is higher in GBM than LGG (Figure 1A), which suggests an important oncogenic role of such receptor in the development of low- and high-grade gliomas. Among gliomas, a higher expression of ER-α could be involved in developing a high-grade glioma (GBM), while lower ER-α expression is associated with LGG development. The decrease or loss of ER expression during tumor development supposes the homeostatic imbalance of the normal functions of ERs in cells. Although this topic is not yet well studied in GBM, in breast cancer, it has been determined in more detail that in the most malignant tumors with multiple metastases, the nuclear factor kappa-light-chain-enhancer of activated B cells (NF-κB) represses ER-α transcription through the enhancer of zeste homolog2 (EZH2), which negatively regulates ER-α transcription [82]. However, NF-kB also improves the recruitment of ER-α to estrogen response elements (EREs) of its target promoters and increases its transcriptional activity [83,84]. The change in ER-α functions may also be due to other factors, such as altered structural conformations that increase interaction with transcriptional coactivators, point mutations that promote active forms of the receptor in the absence of an agonist, or variations by alternative splicing that change receptor transactivation mechanisms [85,86,87]. These data suggest that ER-α expression levels may not always be proportional to its activity. The positive correlation of ER-β expression concerning the GBM malignancy grade observed in this work does not correspond to other studies that find the opposite. However, we must consider that these studies do not specify the brain region of the used healthy tissue. We used expression data from the cerebral cortex, but there are other regions with a higher abundance of ESR2, such as the pons, cerebellum, thalamus, basal ganglia, and hypothalamus. Nevertheless, ESR1 and ESR2 expression on human GBM-derived cell lines were lower compared to healthy astrocytes, reinforcing the hypothesis of the studies above, which establish that the expression of both ER subtypes is inversely proportional to the tumor evolution degree. Different actions have been observed between ER subtypes in GBM. E2 (10 nM) and PPT (1 nM), the selective ER-α agonist, increased the number of cells derived from GBM [45], while treatment with different ER-β-specific agonists decreased GBM cell proliferation [44]. Within the cancer context, a positive correlation has been determined between EMT activation and increased cell proliferation [88,89,90,91]. Previously, our group determined in GBM-derived cells that E2 induces cell growth and the expression of vascular endothelial growth factor (VEGF), epidermal growth factor receptor (EGFR), and cyclin D1 genes, which are involved in cell proliferation. These effects depended on ER-α activation [45]. This previous study represents the primary antecedent of our work. Once we demonstrated that E2 induces cell growth in GBM cells, we decided to investigate the relationship between E2 and canonical cellular processes of EMT activation, such as morphological and actin cytoskeleton organization changes, EMT marker expression, as well as cellular migration and invasion, which represent the main functional consequences of cells that undergo EMT. Additionally, it is worth mentioning that recently Castruccio et al. demonstrated that E2 significantly increases cell proliferation in U87 cells [92], which is consistent with our previously published results [45]. Still, our survival analysis showed that both ER-α and ER-β expression was positively correlated with a poor prognosis. It is interesting to highlight that our results showed that expression of both ER subtypes was enriched in the mesenchymal subtype concerning the other subtypes since this data led us to investigate the role of ERs on the induction of the mesenchymal phenotype.

EMT is a transition process between cellular phenotypes; therefore, it involves cellular morphological changes. To characterize the effects of E2 on EMT-related processes, in this study, we first evaluated changes in the morphology of cells treated with E2. Our results showed that E2 significantly changed the morphology of the four GBM-derived cells, towards an elongated mesenchymal phenotype. Moreover, these changes were correlated with an actin filament rearrangement. E2 regulates the reorganization of actin filaments through phosphorylation of actin-binding proteins such as cofilin and moesin in neurons, fibroblasts, and breast and endometrial cancer cells [93,94,95,96]. During EMT, actin filaments progressively reorganize from thin cortical bundles to thick contractile filaments that withstand stress fibers. The mesenchymal phenotype presents distribution of actin filaments in a front–rear polarization, with a network of short actin filaments branched at the leading edge, and long filaments arranged in different types of fibers behind the leading edge, which are associated with adhesion structures. This arrangement of the mesenchymal actin network allows cells to carry out migratory and invasive processes [97,98,99]. Thus, in this study, we demonstrated that E2 promotes a morphology change in GBM cells towards a mesenchymal phenotype due to the rearrangement of actin filaments, which were assembled in parallel along the ventral surface with long projections toward the leading edge. These changes in cellular morphology and reorganization of actin filaments suggest that E2 provides GBM-derived cells migratory and invasive capabilities.

All molecular events that occur during EMT are spatially and temporally coordinated during the transition. Therefore, the reorganization of the actin filaments is coupled with the expression changes that modify the cellular phenotype. Previously, the EMT definition described a complete transition between two different states, the epithelial and mesenchymal phenotype. Therefore, the primary experimental model for EMT evaluated the decrease in epithelial markers and the increase in mesenchymal markers. However, this perspective has generated extensive debate about the presence of EMT in certain circumstances, such as in cancer progression, which tends to create hybrid epithelial/mesenchymal (E/M) phenotypes that exhibit both epithelial and mesenchymal characteristics in a process known as partial EMT. Hybrid E/M phenotypes in cancer cells present better migratory and invasive capacities, as well as higher resistance to therapy [100,101,102].

We showed that E2 increased the expression of the epithelial marker ZO-1 and the mesenchymal markers N-cadherin and vimentin in U251 and U87 cells. However, the ZO-1 expression regulation was variable between cell lines, since in U251 cells, it only was increased by E2 at the mRNA level at 48 h, while in U87 cells, the expression increased both at the mRNA and protein levels at 48 and 72 h. ZO-1 marker is an adapter protein that binds to multiple components, such as integral proteins of the plasma membrane [103,104], and its presence is essential for assembly of tight and adherent junctions of epithelial cells [105,106]. Therefore, tumor cells from epithelial tissues decrease ZO-1 expression by activating EMT. The implications that ZO-1 expression may have on GBM have not yet been studied. Furthermore, our results showed that the regulation of E2 on ZO-1 expression varies between different human GBM-derived cell lines, possibly due to the different expression profiles among these cells [107,108].

N-cadherin is a transmembrane protein that belongs to the calcium-dependent cell adhesion molecule (CAMs) family that is characteristic of mesenchymal tissue. Increased N-cadherin expression promotes cells to form elongated multicellular chains that migrate faster and more persistently, with a higher proportion of actin stress fibers that provide contractile forces during cell migration [109,110,111]. EMT activation in GBM cells leads to increased expression of N-cadherin, which is associated with increased migratory and invasive capacities. Vimentin is a type III intermediate filament protein that has an essential role in integrity maintaining of mesenchymal cells by providing support and anchorage to organelles, in addition to offering flexibility to cells by stabilizing dynamic interactions of the cytoskeleton during cell migration [112,113,114]. During EMT, there is an extensive change in the composition of intermediate filaments of epithelial cells, which generally express cytokeratin and initiates the expression of vimentin when they differentiate towards the mesenchymal phenotype. Our results showed that E2 promotes the mesenchymal phenotype by increasing the expression of N-cadherin and vimentin both at the mRNA and protein levels in two different human GBM cell lines. Although it was observed that E2 increased the expression of both epithelial and mesenchymal markers, we cannot affirm the induction of a partial EMT, since doing so requires the analysis of more epithelial and mesenchymal markers. However, our results open new perspectives regarding the determination of the status of E2-induced EMT in cells derived from GBM, since estradiol may promote partial EMT. Regardless of the status of EMT induced by E2, our results convincingly showed the acquisition of mesenchymal characteristics in GBM cells by E2 effect, since the changes in the expression of N-cadherin and vimentin are sufficiently forceful in the two cell lines studied.

Taken together, the effects promoted by E2 on GBM cells, such as reorganization of actin filaments as well as increased expression of N-cadherin and vimentin, are related to increased migratory and invasive capacities of U251 and U87 cells. These effects, associated with EMT activation, were replicated with PPT, a selective ER-α agonist that has a 410-fold relative binding affinity for ER-α over ER-β [115], suggesting that the E2 effects on EMT are regulated through ER-α. The latter was verified when using a highly selective ER-α antagonist (MPP, K_i_ = 2.7) [116], which blocked effects produced by E2 on EMT marker expression, as well as the increase of the migratory capacity provided by PPT. Furthermore, neither agonist DNP nor antagonist PHTPP, both selective for ER-β [117], showed significant effects on the expression of EMT markers. Therefore, we conclude that E2 effects observed on the expression of EMT markers are mainly produced by ER-α activation. Much remains to be known about the actions of both ER subtypes during the GBM’s malignant progression. A more in-depth study of the molecular mechanisms of E2 signaling on GBM and its interaction with other signaling factors in specific cellular contexts is necessary for understanding E2 effects on this tumor, which could provide new strategies in GBM treatment.

In this work, we characterized E2-induced EMT in human GBM-derived cells. We found that E2 induces changes in cell morphology through actin filament reorganization and by increased expression of mesenchymal markers. These effects are related to the increased migratory and invasive capacities of GBM cells. Furthermore, E2 effects were mediated by ER-α, since the treatment with its agonist PPT produced similar results to E2, while the treatment with its antagonist MPP blocked these effects. Thus, E2 induces a mesenchymal phenotype through ER-α in cells derived from human GBM.

## Figures and Tables

**Figure 1 cells-09-01930-f001:**
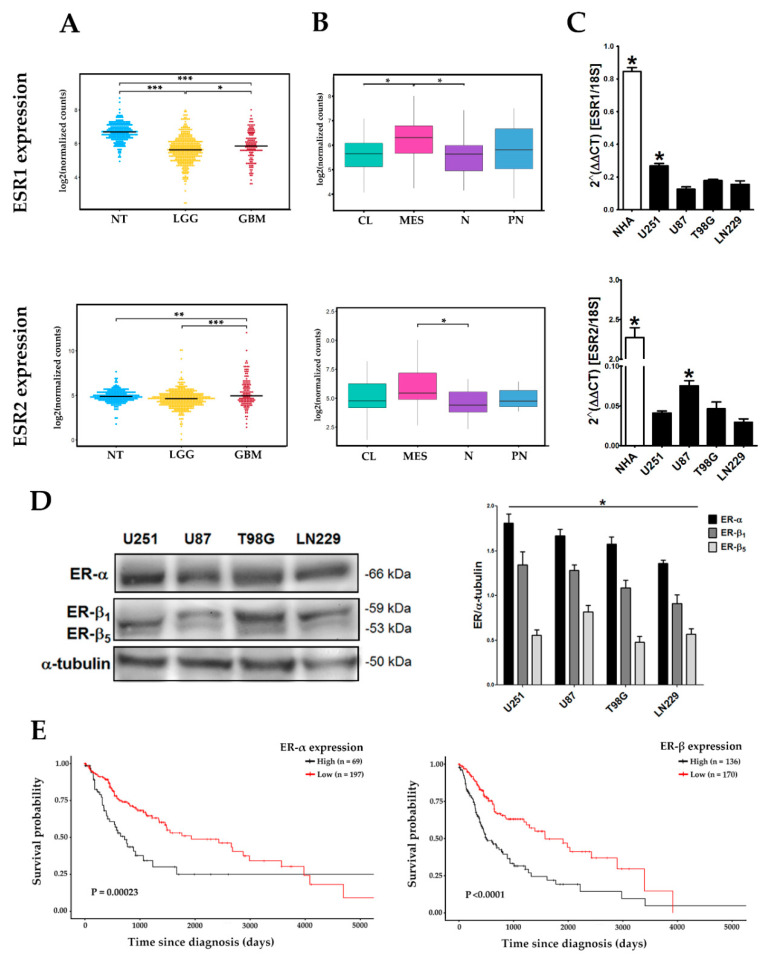
The estrogen receptor-α (ER-α) and estrogen receptor-β (ER-β) subtype gene expressions in human glioblastoma multiforme (GBM). (**A**) Ribonucleic acid sequencing (RNA-Seq) counts obtained from low-grade gliomas (LGG, *n* = 167) and GBM (*n* = 155) projects from The Cancer Genome Atlas (TCGA) and expression profiles obtained from healthy brain cortex samples (normal tissue, NT; *n* = 249) in the GTEx database. LGG includes grade I, II, and III gliomas. * *p* < 0.05; ** *p* < 0.01; *** *p* < 0.001. (**B**) RNA-Seq counts from GBM subtypes: classical (CL), mesenchymal (MES), neural (N), and proneural (PN) obtained from TCGA. * *p* < 0.05. (**C**) RT-qPCR quantified gene expression of estrogen receptor 1/α (ESR1) and estrogen receptor 2/β (ESR2) relative to the reference gene 18S ribosomal RNA (rRNA) using the comparative 2^ΔΔCt^ method in total RNA from normal human astrocyte (NHA) and U251, U87, T98G, and LN229 human GBM-derived cells. Both receptor subtypes were less expressed in GBM cells than in NHAs. * *p* < 0.05 vs. all other groups; mean ± standard error of the mean (SEM), *n* = 3. (**D**) ER-α and ER-β content analyzed by Western blot using α-tubulin as load control. The two main isoforms of ER-β expressed in GBM are shown: ER-β1 and ER-β5. Representative blot image and the corresponding densitometric analysis for ERα and ERβ expression in human GBM-derived cells. * *p* < 0.05 ER-α vs. ER-β and ER-β1 vs. ER-β5; mean ± SEM, *n* = 3. (**E**) Survival analysis for ER-α and ER-β expression in GBM using TCGA data.

**Figure 2 cells-09-01930-f002:**
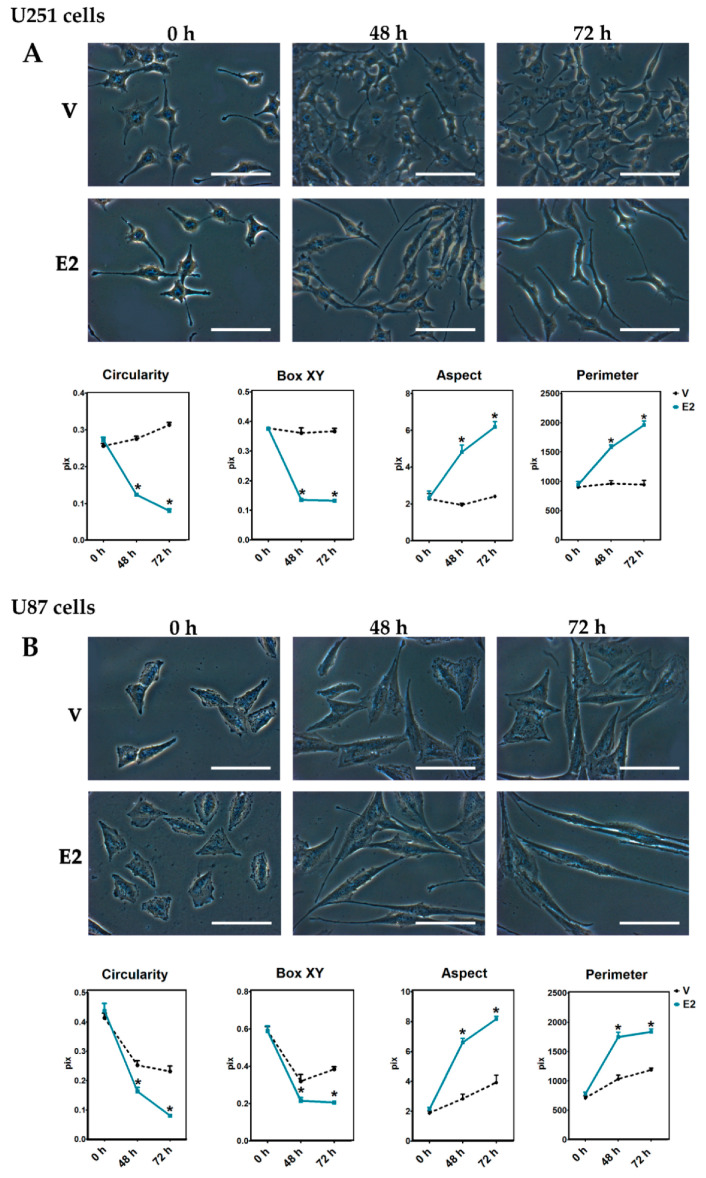
17β-Estradiol (E2)-induced morphological changes in human GBM-derived cells. (**A**) U251 and (**B**) U87 cells observed by phase-contrast microscopy with a magnification of 400× at 0, 48, and 72 h after adding 17β-estradiol (E2, 10 nM) and the vehicle (V, 0.01% cyclodextrin). Magnification white bar = 100 µm. Plots represent the quantification of the geometric parameters (circularity, box XY, aspect, perimeter) in this study. Results are expressed as the mean ± standard error of the mean (SEM); *n* = 3; * *p* < 0.05 vs. V.

**Figure 3 cells-09-01930-f003:**
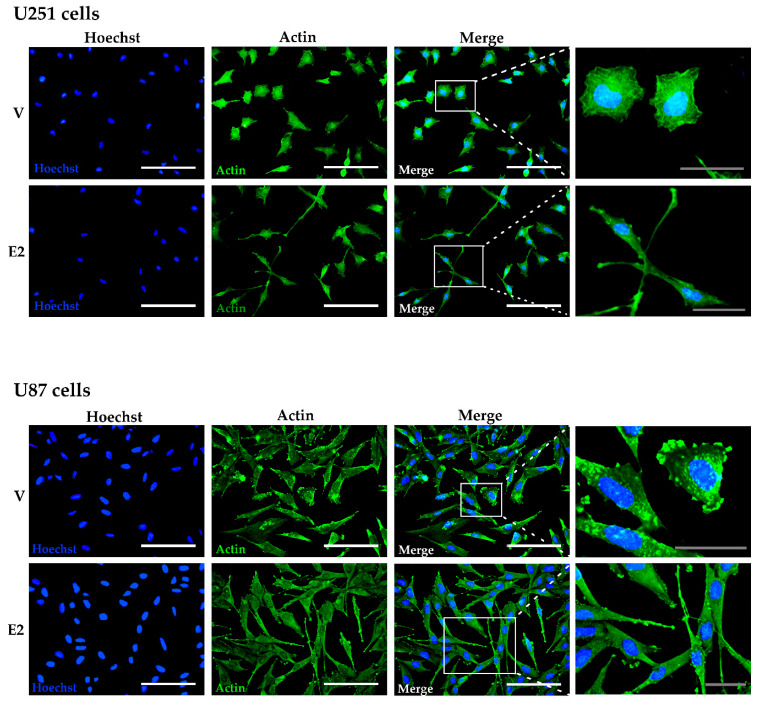
E2 rearranged the actin cytoskeleton of human GBM-derived cells. Actin immunostaining in U251 and U87 cells treated with 17β-estradiol (E2, 10 nM) and vehicle (V, 0.01% cyclodextrin) for 48 h. Representative images captured under a fluorescence microscope at a magnification of 400×. Magnification white bar = 100 µm and gray bar = 30 µm.

**Figure 4 cells-09-01930-f004:**
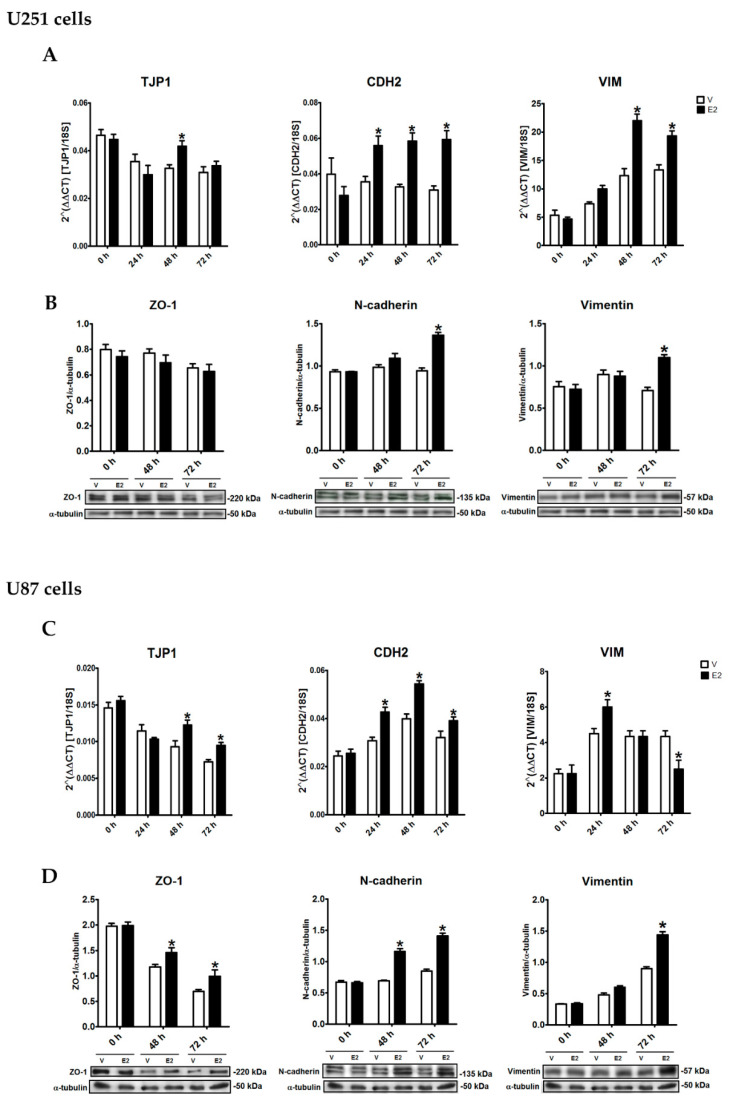
E2-regulated epithelial-to-mesenchymal transition (EMT) marker expression of human GBM-derived cells. (**A**,**B**) U251 and (**C**,**D**) U87 cells were treated with 17β-estradiol (E2, 10 nM) and vehicle (V, 0.01% cyclodextrin) for 24, 48, and 72 h. (**A**,**C**) Epithelial gene (tight junction protein 1 (TJP1)) and mesenchymal genes (vimentin (VIM) and cadherin-2/N-cadherin (CDH2)) expression was quantified by RT-qPCR using the comparative method 2^ΔΔCt^ concerning the reference gene 18S rRNA. (**B**,**D**) Zonula occludens 1 (ZO-1), N-cadherin, and vimentin content was determined by Western blot. Densitometric analysis of EMT marker expression with their respective representative bands using α-tubulin as a load control showed that E2 increased EMT marker expression with different temporal dynamics. Results are expressed as the mean ± standard error of the mean (SEM); *n* = 3; * *p* < 0.05 vs. V.

**Figure 5 cells-09-01930-f005:**
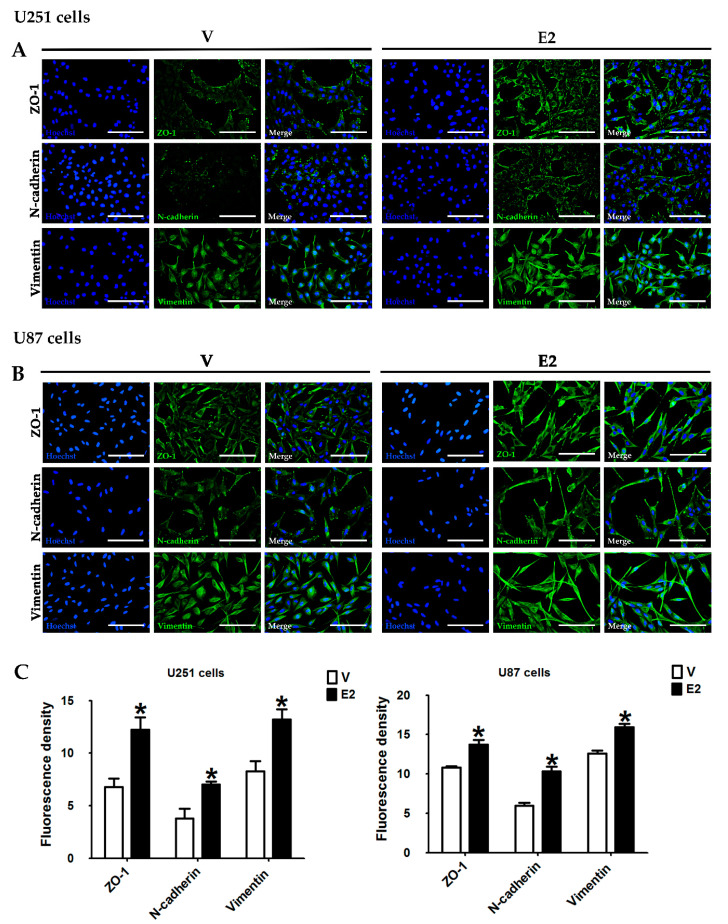
E2 modified EMT marker distribution and immunoreactivity in human GBM-derived cells. ZO-1, N-cadherin, and vimentin immunostaining in (**A**) U251 and (**B**) U87 cells treated with 17β-estradiol (E2, 10 nM) and vehicle (V, 0.01% cyclodextrin) for 48 h. Representative images were captured under a fluorescence microscope at a magnification of 400×. (**C**) EMT marker expression measured as a fluorescence density. Results are expressed as the mean ± standard error of the mean (SEM); *n* = 3; * *p* < 0.05 vs. V.

**Figure 6 cells-09-01930-f006:**
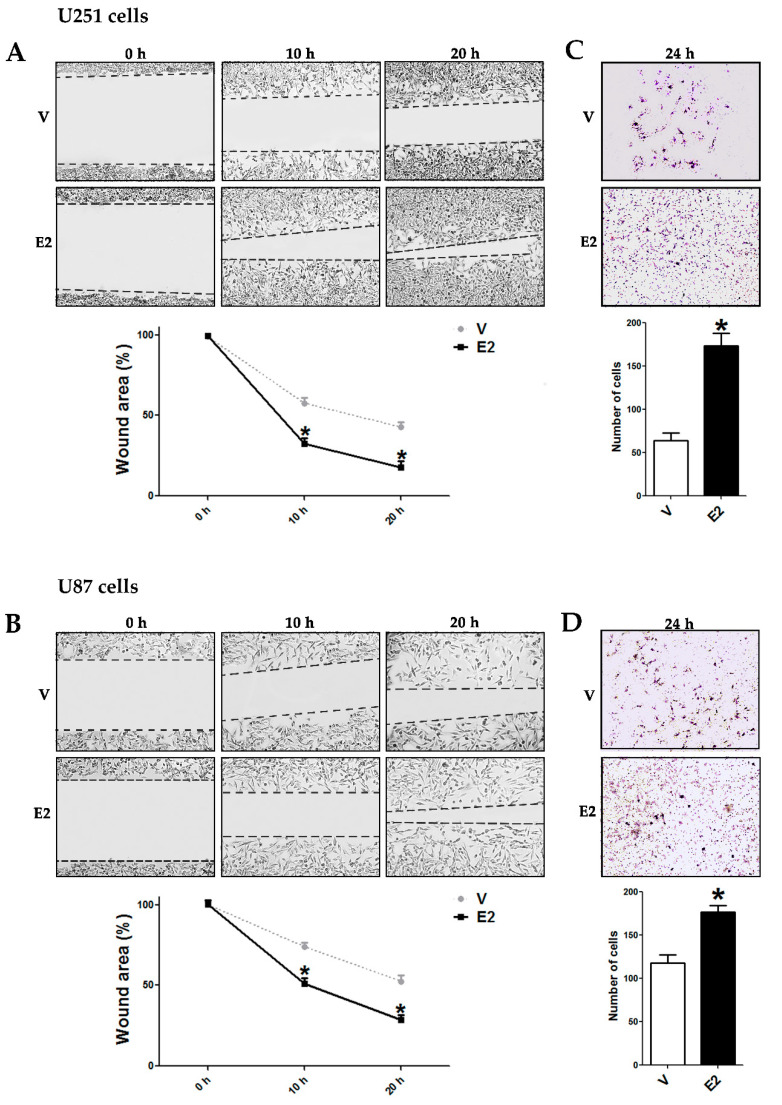
E2 increased migration and invasion of human GBM-derived cells. (**A**,**B**) Wound healing assays were performed in U251 and U87 cells treated with 17β-estradiol (E2, 10 nM) and vehicle (V, cyclodextrin 0.01%). Representative images of wound closure at 0, 10, and 20 h and quantification of the wound area are shown. (**C**,**D**) Transwell assays were carried out in both cell lines. Quantification of cells staining with 0.1% crystal violet dye shows the number of invasive cells. Results are expressed as the mean ± standard error of the mean (SEM); *n* = 3; * *p* < 0.05 vs. V.

**Figure 7 cells-09-01930-f007:**
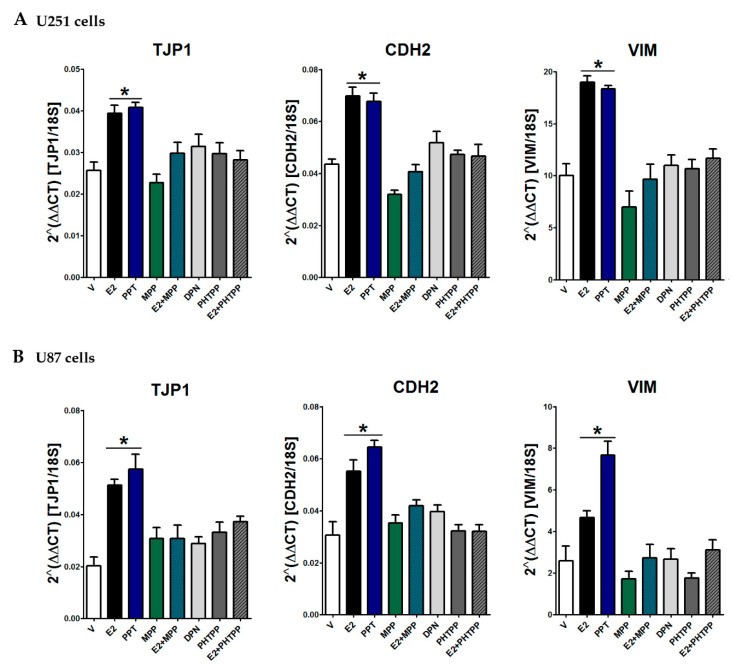
Effect of selective ER-α and ER-β agonists and antagonists on the EMT marker expression in GBM cells. U251 (**A**) and U87 (**B**) cells were treated with vehicle (V, 0.01% cyclodextrin + 0.01% DMSO), 17β-estradiol (E2, 10 nM), 4,4’,4’’-(4-propyl-[1H]-pyrazole-1,3,5-triyl)trisphenol (PPT, 10 nM, selective ER-α agonist), methyl-piperidino-pyrazole (MPP, 1 µM, selective ER-α antagonist), E2 + MPP, diarylpropionitrile (DPN, 10 nM, selective ER-β agonist), 4-[2-phenyl-5,7-bis (trifluoromethyl)pyrazole[1,5-a]pyrimidin-3-yl]phenol (PHTPP, 1 µM, selective ER-β antagonist), and E2 + PHTPP for 48 h. Epithelial gene (TJP1) and mesenchymal gene (VIM and CDH2) expression were quantified by RT-qPCR using the comparative method 2^ΔΔCt^ concerning the reference gene 18S rRNA. Results are expressed as the mean ± standard error of the mean (SEM); *n* = 3; * *p* < 0.05 vs. all other groups.

**Figure 8 cells-09-01930-f008:**
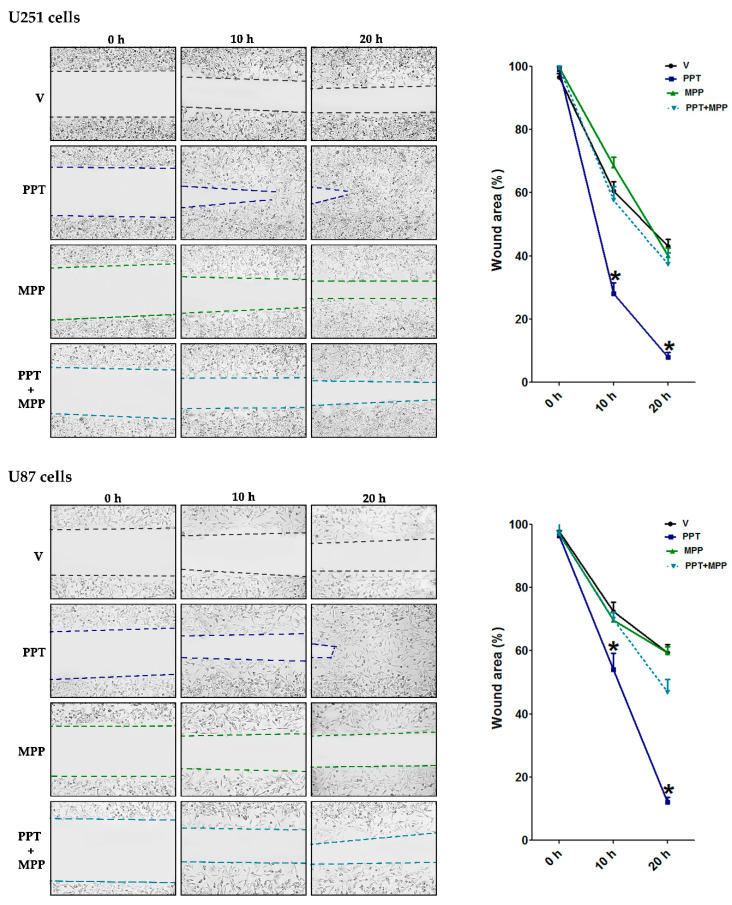
Effect of the selective ER-α agonist and antagonist on the migration of GBM cells. (**A**) U251 and (**B**) U87 cells that were treated for 20 h, with PPT (10 nM, selective ER-α agonist), MPP (1 μM, selective ER-α antagonist), PPT + MPP, and vehicle (V, DMSO 0.01%). Wound healing assays determined the migratory capacity of the U251 and U87 cells. Representative images show wound closure at 0, 10, and 20 h and quantification of the wound area. Results are expressed as the mean ± standard error of the mean (SEM); *n* = 3; * *p* < 0.05 vs. all other groups.

**Table 1 cells-09-01930-t001:** Geometric parameters of Image-Pro software.

Parameter	Description	Image
Area	The area included in the polygon that defines the figure contour	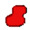
Axis major	Major axis length of an imaginary ellipse surrounding figure	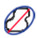
Axis minor	Minor axis length of an imaginary ellipse surrounding figure	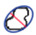
Aspect	The ratio between the major and minor axis of an ellipse	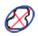
Bound box height	Bounding box height of the figure	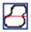
Bound box width	Bounding box width of the figure	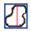
Box XY	The ratio of width to height of bounding box	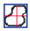
Circularity	The ratio of figure area to the diameter of a circle around it	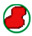
Perimeter	Length of the region surrounding the figure	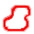

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
