# Peer review of "Estradiol Induces Epithelial to Mesenchymal Transition of Human Glioblastoma Cells"

_cells, 2020, doi:10.3390/cells9091930_

Round 1

Reviewer 1 Report

This manuscript shows that ER alpha receptor is a critical role of survival in patients with GBM because estradiol induces epithelial-mesenchymal transition to be a cell plasticity in GBM.
Authors present the data that estradiol, including ER alpha agonist but not ER beta agonist, changes in cell morphology, upregulates the expression of EMT markers, facilitates cell migration and invasion in human GBM-derived cell lines using qPCR, western blot, and several assays.
Authors, also, show that ER alpha antagonist, but not ER beta antagonist, blocks estradiol-induced epithelial-mesenchymal transition.
This manuscript is very interesting and has many suggestions.

Minor
Estradiol enhanced malignancy involving cell migration and invasion.
However, I missed the data that estradiol effects cell proliferation.
I believe the data of cell proliferation may improve this manuscript.

Reviewer 2 Report

Hernandez-Vega et al. show that estradiol (E2) induces epithelial-mesenchymal transition (EMT) in human glioblastoma (GBM) cells. Estradiol induced the morphological changes associated with EMT and reorganized actin filaments. Authors show that E2 regulated the expressions of epithelial and mesenchymal markers, promoted the migratory and invasive capacity of GBM cells. Authors then determined which estrogen receptor mediated the effects of E2 and found that ER-a agonist PPT increased mRNA levels of both epithelial and mesenchymal markers. ER-b agonist and antagonist did not have any effect on the expression of EMT markers. Finally, the authors show that ER-a agonist PPT increased the migration capacity of GBM cells and this effect was blocked by ER-a antagonist MPP. Authors therefore conclude that the effects of E2 on induction of EMT in GBM cells are mediated by ER-a and not ER-b.

Authors provide evidence for induction of EMT by estrogen through ER-a. However, it is unclear if authors conclude this as a complete EMT or a partial EMT as expressions of epithelial as well as mesenchymal markers are upregulated by estrogen. Addressing following points is necessary to further support the occurrence of EMT and to strengthen the findings.

  1. It is unclear and misleading what authors are trying to explain about the expressions of ESR1 (ER-α) and ESR2 (ER-β) in Figure 1. It shows that ESR1 has low expression in GBM tissue in TCGA dataset compared to healthy tissue and also in GBM cells compared to normal human astrocytes (NHA). However, authors also show that high ER-α expression is associated with lower survival in Figure 1E. Does this mean that astrocytes and normal tissues, which have higher ER-α expression than GBM cells have malignant role? If ER-α is mediating the effects of E2 in inducing EMT and more aggressive phenotype of GBM cells by increasing migration and invasion, and is associated with lower survival, then, its expression is expected to be higher in GBM tissue and cells compared to normal tissue and normal astrocytes. These results are confusing. Authors have tried to explain these in the discussion section, but it’s not entirely clear. Additional explanation with detailed interpretation of these results should be included.
  2. E2 and ER-α agonist increase the mRNA and protein levels of epithelial as well as mesenchymal markers. This is unexpected as in traditional EMT, the expression of epithelial markers decreases, whereas the expression of mesenchymal markers increases. However, as authors have indicated in the discussion section about hybrid EMT, this concept should be explicitly mentioned and explained in the results, which otherwise, can be confusing to the readers. Also, additional evidence using more markers should be included to convincingly demonstrate the occurrence of hybrid EMT, if that is what authors believe is happening.
  3. As mentioned above about the inclusion of additional epithelial and mesenchymal markers, E-cadherin is very well-known epithelial marker and effects of E2 and its agonists, antagonists should be tested on E-cadherin mRNA and protein levels. Additional epithelial markers such as cytokeratin, MUC-1 and mesenchymal markers such as α-SMA, Snail, fibronectin should be analyzed to confirm that E2 and ER- α agonist increase expression of epithelial as well as mesenchymal markers as current evidence using a single epithelial marker and two mesenchymal markers is not convincing, especially due to the case of hybrid EMT.
  4. Authors show that ER-a agonist PPT agonist increased mRNA levels of mesenchymal markers, CDH2, VIM, and also, epithelial marker TJP1. ER-a antagonist, MPP, is expected to have opposite effects and decrease the mRNA levels of these markers as authors argue that ER-a is mediating the effects of E2. This is not the case and there is no change in expressions of any markers after MPP treatment as shown in Figures 7 A, B. This should be explained.
  5. Similar to point 4 above, ER-a antagonist MPP does not have any effect by itself on migration of cells in wound healing assay, but only blocks the effect of the agonist. As agonist PPT increases the wound closure rate of cells by itself, antagonist MPP is expected to decrease this rate.
  6. Does ER-a agonist increase the invasiveness of GBM cells and antagonist decrease the invasiveness? Authors have not tested the effects of any agonist or antagonist on invasion of cells. This analysis should be performed and included in the manuscript.

Round 2

Reviewer 2 Report

In the revised version of the manuscript, authors have addressed my concerns by including information as required throughout the manuscript.